# Recombinant Human HPS Protects Mice and Nonhuman Primates from Acute Liver Injury

**DOI:** 10.3390/ijms222312886

**Published:** 2021-11-28

**Authors:** Yang Yang, Huali Zhai, Yue Wan, Xiaofang Wang, Hui Chen, Lihou Dong, Taoyun Liu, Guifang Dou, Chutse Wu, Miao Yu

**Affiliations:** 1Department of Pharmaceutical Engineering, School of Chemical Engineering and Technology, Tianjin University, Tianjin 300072, China; yang01261989@163.com (Y.Y.); zhl0409@126.com (H.Z.); 2State Key Laboratory of Proteomics, Beijing Proteome Research Center, National Center for Protein Sciences (Beijing), Institute of Lifeomics, Beijing 102206, China; 13572956203@163.com (Y.W.); wangxiaofang8515@163.com (X.W.); chenhui19811212@sina.com (H.C.); dlh9957001@sina.com (L.D.); 3School of Basic Medical Sciences, An Hui Medical University, Hefei 230032, China; 4Institute of Life Sciences, He Bei University, Baoding 071002, China; 5Department of Pharmacology and Toxicology, Beijing Institute of Radiation Medicine, Beijing 100850, China; liutaoyun@163.com (T.L.); dougf@bmi.ac.cn (G.D.)

**Keywords:** hepassocin, hepatocyte death, acute liver injury

## Abstract

Acute liver injury shares a common feature of hepatocytes death, immune system disorders, and cellular stress. Hepassocin (HPS) is a hepatokine that has ability to promote hepatocytes proliferation and to protect rats from D-galactose (D-Gal)- or carbon tetrachloride (CCl_4_)-induced liver injury by stimulating hepatocytes proliferation and preventing the high mortality rate, hepatocyte death, and hepatic inflammation. In this paper, we generated a pharmaceutical-grade recombinant human HPS using mammalian cells expression system and evaluated the effects of HPS administration on the pathogenesis of acute liver injury in monkey and mice. In the model mice of D-galactosamine (D-GalN) plus lipopolysaccharide (LPS)-induced liver injury, HPS treatment significantly reduced hepatocyte death and inflammation response, and consequently attenuated the development of acute liver failure. In the model monkey of D-GalN-induced liver injury, HPS administration promoted hepatocytes proliferation, prevented hepatocyte apoptosis and oxidation stress, and resulted in amelioration of liver injury. Furthermore, the primary pharmacokinetic study showed natural HPS possesses favorable pharmacokinetics; the acute toxicity study indicated no significant changes in behavioral, clinical, or histopathological parameters of HPS-treated mice, implying the clinical potential of HPS. Our results suggest that exogenous HPS has protective effects on acute liver injury in both mice and monkeys. HPS or HPS analogues and mimetics may provide novel drugs for the treatment of acute liver injury.

## 1. Introduction

Liver injury is a common disease that seriously threatens the life and health of patients. It is caused by numerous factors, such as virus infection, toxic compounds, metabolic diseases, and genetic disorders. Acute liver failure (ALF) is characterized by a fast-evolving hepatic dysfunction associated with massive hepatocyte apoptosis, hemorrhagic necrosis, inflammation, coagulopathy, sepsis, and multi-organ failure, which causes over 60% mortality if liver transplantation is not provided [1]. Despite different causes of ALF, hepatocytes death, immune system disorders, and mitochondrial dysfunction are certain pathological features [2,3]. Strategies targeting these pathogenic processes are efficacious therapeutic approaches for the treatment of human liver diseases.

HPS, also named fibrinogen-like protein 1 (FGL1) or hepatocyte-derived fibrinogen-related protein 1 (HFREP1), belongs to a fibrinogen family, and has been implicated as a mitogen for hepatocytes in a mitogen-activated protein kinase (MAPK)-dependent manner [4,5]. HPS is expressed mainly in the liver, while it is also low-level expressed in the adipose and the pancreas [4,6]. Serum and hepatic HPS levels are induced in different kinds of stress conditions including partial hepatectomy [5], fatty acids expose [7], radiation [8], endoplasmic reticulum (ER) stress [9], and acute inflammation [10], indicating HPS is an acute phase reactant, and its upregulation may be an important adaptive response against liver injury. Plasma HPS concentrations were elevated in patients with nonalcoholic fatty liver disease (NAFLD) [11], diabetes [12], and hyperglycemic crisis [13], while decreased significantly in subjects with hyperglycemic crisis after standard treatment accompanied by improved hepatic functions [13]. Administration of HPS to rats after D-galactose (D-Gal) and carbon tetrachloride (CCl_4_) treatment significantly inhibited the development of ALF by preventing the high mortality rate and improving liver histology [5]. HPS has the ability of anti-apoptosis by inhibiting the upregulation of toxin-induced proapoptotic factors (BCL2-associated X protein (BAX), cleaved caspase 9, and increasing the expression levels of anti-apoptotic factors (B cell leukemia/lymphoma 2 (BCL-2), BCL2-like 1 (BCL-XL)) [5]. Mesenchymal stem cell (MSC) conditioned medium-attenuated CCl_4_-induced early apoptosis in hepatocytes through activation of HPS [14]. HPS increased antioxidative stress proteins to protect hepatocytes from high glucose-induced glucotoxicity, as indicated by improved hepatic functions and histologic changes [13]. Bone marrow-derived mesenchymal stem cells (BMSCs) have been demonstrated to exert extensive therapeutic effects on acute liver injury, such as inhibition of hepatocyte apoptosis and acceleration of liver regeneration [15]. HPS was reported to contribute to the therapeutic effects of BMSCs [15]. Additionally, knockdown of endogenous HPS in vivo enhanced the liver injury induced by D-Gal, suggesting HPS may be one of the endogenous physiological protectors for hepatocytes [5]. HPS has been found to be reduced or undetectable in most hepatocellular carcinoma (HCC) specimens at both the RNA and protein level [16,17]. Mice lacking HPS accelerated the development of HCC following tumor induction [18]. These data indicated that HPS might possess growth suppression activity in HCC. Based on these observations, HPS is thought to be a candidate drug for ALF. Further studies to elucidate the pharmacological roles of HPS in detail would provide important insights for the use of HPS as a therapeutic drug to treat liver diseases.

To further confirm the hepatoprotective role of HPS, we here generated a pharmaceutical-grade recombinant human HPS, and evaluated the effects of HPS administration on the pathogenesis of acute liver injury in mice and monkeys. Our results showed that HPS administration significantly alleviated liver injury in both animal models, which provides a novel basis for the application of HPS in the treatment of liver diseases. In addition, the primary pharmacokinetic and safety evaluation in animals suggest the clinical potential of HPS.

## 2. Results

### 2.1. Expression and Characterization of Recombinant Human HPS Produced by CHO Cells

CHO cell clones with high expression of HPS, the productivity of which achieved 300 ng/mL, were selected through pressure screen, and used for batch culture. After purification by hydrophobic chromatography and ion exchange, the purity of HPS reached 95% (Figure 1A and Table 1). Western blot analysis using an anti-human HPS antibody showed that the purified HPS is at the expected size of 34 and 68 kDa under reducing and non-reducing conditions, respectively (Figure 1B). The N-terminal amino acid sequence of the HPS purified from the culture medium of cells is LED[C]AQEQMRLRAQV, which is coincided with that deduced from the nucleotide sequence of the human HPS cDNA. Other quality results such as host cell protein (HCP), host cell DNA (HCD), and endotoxin are summarized in Table 1.

The biologic activity of the HPS was analyzed by in vitro and in vivo assays as previous reports [5]. Treatment with HPS stimulated hepatocyte proliferation in a dose-dependent manner with increased the phosphorylation of ERK1/2 (Figure 1C,D). Administration of HPS showed protective effects against CCl_4_-induced liver injury and increased survival rates in mice (Figure 1E–G).

### 2.2. HPS Treatment Attenuates D-Galactosamine plus Lipopolysaccharide (D-Galn/LPS)-Induced Liver Injury in Mice

Low doses of lipopolysaccharide (LPS) challenge combined with hepatocyte-specific transcriptional inhibitor D-galactosamine (D-GalN) leads to ALF, which is frequently used as an animal model that closely represents ALF in clinic [19]. We thus examined the effects of HPS on D-GalN/LPS-induced acute liver injury. Injection of a lethal dose of D-GalN/LPS caused 90% mortality in PBS treated mice within 72 h (h), but HPS treatment significantly protected mice from this challenge in a dose-dependent manner. About 75% of mice survived to 168 h in the high dose HPS treatment group (Figure 2A). After injection of a sublethal dose of D-GalN/LPS, a significant increase in serum alanine aminotransferase (ALT) and aspartate aminotransferase (AST) was observed; however, the administration of HPS significantly suppressed the increase of ALT and AST in a dose-dependent manner (Figure 2B). Histological examination revealed serious necrosis, apoptosis, and inflammatory areas in the liver after D-GalN/LPS exposure, while HPS treatment potently suppressed these pathological processes (Figure 2C).

### 2.3. HPS Treatment Inhibits Pro-Inflammatory Cytokine Production and Hepatic Oxidative Stress in Mice after D-Galn/LPS Injection

We next investigated the effect of HPS treatment on the production of pro-inflammatory cytokines induced by D-GalN/LPS administration. D-GalN/LPS administration induced rapid increases in serum pro-inflammatory cytokines, such as tumor necrosis factor alpha (TNF-α), interferon gamma (IFNγ), interleukin 6 (IL-6), and monocyte-chemoattractant protein-1 (MCP1), but the increase was suppressed by HPS treatment (Figure 3A). D-GalN/LPS administration also upregulated mRNA expression of several major inflammatory mediators in liver, including *TNF-α*, *IL-6*, *IFNγ*, and interleukin 1 beta (*IL-1β*); however, HPS treatment decreased the levels of these mediators (Figure 3B).

Acute hepatitis induced by D-GalN/LPS is partly mediated by oxidative stress [19]. Therefore, we examined whether HPS affects D-GalN/LPS-induced oxidative stress by estimating protein carbonylation and malondialdehyde (MDA), which represent oxidative protein damage and lipid peroxidation. D-GalN/LPS significantly induced the carbonylation of hepatic proteins in mice, while HPS treatment inhibited the carbonylation of hepatic proteins in a dose-dependent manner (Figure 3C). HPS treatment also decreased the D-GalN/LPS induced accumulation of MDA in liver tissue of mice in a dose-dependent manner (Figure 3D). Moreover, HPS treatment significantly reduced hepatic reactive oxygen species (ROS) levels and dihydroethidium (DHE)-positive staining and up-regulated superoxide dismutase 2 (*SOD2*) and catalase (*CAT*) mRNA expressions (Figure 3E-3G). These results suggest that HPS attenuates D-GalN/LPS-induced hepatic oxidative stress in mice.

### 2.4. Peripheral Delivery of Recombinant Human HPS Prevents Acute Liver Injury in Nonhuman Primates

Cynomolgus monkeys were intravenously administered with D-GalN (300 mg/kg), and randomly divided into HPS-treated group and saline-treated group. Following the toxin injection, the saline-treated monkeys showed significantly poor appetite and vomiting, torpidity, drowsiness, and mental indifference. In contrast, the monkeys that received HPS consistently maintained better physical and mental conditions. D-GalN administration resulted in increased serum ALT and AST, total bilirubin (TBIL), direct bilirubin (DBIL), and alkaline phosphatase (ALP) levels, suggesting that D-GalN could induce liver injury in monkey, which was consistent with previous reports [20]. While HPS administration significantly decreased the levels of these markers (Figure 4A), which indicates that exogenous HPS could protect monkeys against D-GalN-induced liver injury. Histology disclosed apparently lower parenchymal swelling and necrosis in HPS treated animals at 96h after toxin injection (Figure 4B). Consistently, animals treated with HPS displayed a small amount of hepatocyte apoptosis detected by terminal deoxynucleotidyl transferase-mediated dUTP nick end labeling (TUNEL) (Figure 4C). HPS-treated monkeys exhibited lower levels of inflammatory cytokines than the control animals including TNF-α, IL-6, and IFNγ at 96h after D-GalN injection (Figure 4D). The levels of serum and hepatic 4-hydroxynonenal (4-HNE) positive cells were decreased in HPS-treated animals, indicating that HPS treatment reduced D-GalN-induced hepatic oxidative stress in monkeys (Figure 4E,F). Moreover, the hepatocyte proliferation was markedly higher in HPS-treated animals compared to control mice at 96 h after D-GalN challenge, as measured by proliferating cell nuclear antigen (PCNA) staining (Figure 4G).

### 2.5. The Pharmacokinetic Evaluation of HPS in Rats

We evaluated the pharmacokinetic properties of HPS in SD rats administered a single intravenous or subcutaneous dose (3.5 mg/kg). Mean plasma HPS concentration vs. time curve after administration of HPS was summarized in Figure 5A, respectively. The corresponding pharmacokinetic parameters are listed in Table 2 using the scientific application package Kinetica VR version 4.1.1. By intravenous injection, the area under the plasma concentration-time curve was 27.79 ± 2.66 (h·µg)/mL with a maximal plasma concentration (Cmax) of 47.78 ± 5.01 µg/mL and a half-life of 1.57 h. The rapid absorption of HPS was observed after subcutaneous injection, producing peak concentration of 33.93 ± 2.06 µg/mL at 15 min. Subcutaneous bioavailability of HPS was 100%.

HPS was labeled with ^125^I and injected intravenously into SD rats to analyze its tissue distribution. As shown in Figure 5B, HPS is more likely to be distributed in tissues with rich blood perfusion, such as lung, liver, spleen, and kidney. Due to the low radioactivity of ^125^I in the brain, HPS may not easily penetrate the blood–brain barrier. The results of fecal and urine excretion suggest that HPS is mainly excreted through the kidneys (Figure 5C).

### 2.6. Acute Toxicity Study

We preliminarily evaluated the safety of HPS in mice after administration of a high dose of HPS (20 times the effective dose). There was an absence of mortality during the whole observational period and no significant differences in the body weights (Figure 6A), feed and water consumption (data not shown) were observed in HPS-treated group as compared to control group. Both HPS-treated mice and control mice have shown sensitivity and normal behavior to different kinds of stimuli (data not shown). Animal feces were in regular form without any change in color (data not shown). Biochemical blood analysis showed that administration of HPS did not alter any of biochemical parameters such as ALT, AST, urea nitrogen (BUN), creatinine (CREA), and creatine kinase (CK) (Figure 6B). HPS-treated mice displayed normal cellularity of peripheral blood (PB) and constitution of leukocytes in PB (Figure 6C). No significant alterations in the relative weight of heart, liver, lung, kidneys, and spleen were found in HPS-treated animals (data not shown). Moreover, histopathological studies revealed that HPS treatment did not cause microscopic abnormality in liver (Figure 6D), heart, lung, kidneys, and spleen (data not shown). Collectively, our data suggest that the mice are well tolerated to HPS at the dose of 20 mg/kg during the study period. Previous studies have reported that overexpression of HPS increases lipid accumulation in liver [11,12]. However, we observed no increase in hepatic lipid accumulation in wild type (WT) and *Lep^ob^*/*Lep^ob^* (ob/ob) mice after 1 week of HPS (1mg/kg body weight) treatment (Figure 7A,B).

## 3. Discussion

In this paper, we elucidated the importance of HPS in hepatoprotection using a pharmaceutical-grade recombinant human HPS. In previous studies, we generated human HPS in *E. coli* and showed its bioactivity by stimulation of hepatocyte proliferation in vitro and treatment of acute liver injury in rats [5]. HPS expressed in *E. coli* resulted in inclusion bodies and low yield [5]. Although the soluble expression and purification of recombinant HPS have been achieved in *E. coli* by fusion technology [21], there is still a need to facilitate correct protein folding, enhance solubility, and alleviate proteolytic degradation, as well as increase the activity of expressed HPS. To prepare large-scale production of HPS, we here successfully developed a procedure to express soluble and functional recombinant human HPS in CHO-DG44 cells. In line with previous results, HPS generated in this study stimulated hepatocyte proliferation, triggered ERK1/2 phosphorylation, and protected mice from CCl_4_ induced liver injury. The program described here greatly increased the production efficiency of functional HPS, which may be useful for future scale-up and possible clinical development.

Co-administration of D-GalN/LPS-induced hepatitis is a well-established model similar to ALF in the clinical setting [19]. The advantage of this model is that D-GalN/LPS is not directly hepatotoxic, which induce liver injury through the induction of hepatic inflammation. Pathogen associated molecular patterns (PAMPs) such as LPS has been linked with different kinds of ALF or ALF on chronic liver failure background [19]. We demonstrated that administration of HPS significantly alleviated D-GalN/LPS-induced acute liver injury in mice by preventing the high mortality rate and improving liver histology including necrosis, apoptosis, and inflammatory. These results confirm that HPS has a role in protection against Toll-like receptors-mediated liver injury. Importantly, we found that D-GalN/LPS-induced hepatic oxidative stress was significantly attenuated by HPS administration, as evidenced by reduced the levels of ROS, protein carbonylation and MDA accumulation, suggesting that HPS ameliorates acute hepatitis induced by D-GalN/LPS at least in partial by inhibiting oxidative stress. We further demonstrated that administration of HPS significantly alleviated D-GalN-induced acute liver injury in nonhuman primates, as evidenced by reduced serum ALT activity attenuated hepatocytes apoptosis, increased hepatocytes proliferation, and improved liver histology. In addition, HPS treatment also inhibited hepatic oxidative stress in D-GalN-treated nonhuman primates. These findings agree with previous reports claiming an important role of HPS in rats’ liver injury models [5], and highlight HPS delivery may be a potential therapeutic strategy for the treatment of acute liver injury in humans, mostly in patients who would need long-term treatment with potentially hepatotoxic drugs. Based on previous work and data presented here, we conclude that the improvement of liver injury by HPS is mainly relevant from four perspectives: Firstly, by stimulating hepatocyte regeneration through activation of MAPK/ERK pathway [4]; secondly, by attenuating hepatocytes apoptosis through decreased proapoptotic protein Bax and elevated antiapoptotic protein Bcl-2 [5,15]; thirdly, by reducing oxidative stress response; fourthly, by inhibiting inflammation. Previous studies showed that HPS may specifically target hepatocytes; however, recent studies revealed that HPS has a negative regulatory effect on immune cells through binding lymphocyte-activation gene 3 (LAG3) [22]. We found that HPS treatment significantly reduced proinflammatory cytokine production in D-GalN/LPS challenged animals, suggesting HPS may have a role to modulate the hepatic immune microenvironment.

Poor pharmacokinetics properties are an important factor limiting the use of proteins or peptides for therapeutic purposes. The pharmacokinetic test showed that HPS was rapidly cleared from blood circulation through renal filtration, with a half-life of about 1.57 h. After subcutaneous injection, HPS was rapidly absorbed, with a peak at 0.25 h. Subcutaneous bioavailability of HPS was 100%. These data suggest that natural HPS possesses favorable pharmacokinetics. On the other hand, we noted that the large dose of HPS seems to be required for its in vivo functions, it may be due to natural HPS has a short half-life, therefore, maintaining HPS levels high enough may be necessary to achieve therapeutic effects. Development of HPS analogues and mimetics with improved pharmacokinetic profiles and potency may be suitable for future clinical efficacy.

Most natural HPS in plasma were found to bound to the fibrin matrix during clot formation [23]. Although the role of HPS in clot formation is unknown, the potential procoagulant effects may limit the therapeutic applications of HPS. In addition, HPS has been reported to regulate metabolism and inhibit immune response. HPS deficient mice displayed obesity, abnormal plasma lipid profiles, structural defects in adipose tissues, and developed ulcerative dermatitis in aged mice [6,22]. Preliminary data from the safety test showed that injection of a high dose of HPS caused no obvious adverse effect or impairment on the health condition of experimental animals. Specifically, HPS treated mice displayed normal weight, cellularity of peripheral blood and hepatic lipid accumulation. These data implied the clinical potential of HPS.

In conclusion, we developed a procedure to produce functional recombinant human HPS. The generated HPS can protect monkeys and mice from acute liver injury. Moreover, primary pharmacokinetic test and acute toxicity study implied the clinical potential of HPS. These data suggest that HPS may be a potential therapeutic drug for the treatment acute liver injury.

## 4. Materials and Methods

### 4.1. Expression and Purification of Human HPS

The human full-length HPS was amplified by PCR from the human liver cDNA and subcloned into the pGN-M vector (GenScript, Nanjing, Jiangsu, China), which contained mouse dihydrofolate reductase (DHFR) gene. After linearization with Pvu I, the plasmid encoding human HPS was transiently transfected into CHO-DG44 cells. Then the transfected cells were screened using limited dilution method and amplified by increasing the concentration of methotrexate (MTX) by gradient in the medium. After 4 rounds of pressure screening, clones with high expression of HPS were selected. The HPS stable cell line was expanded and inoculated into 10-L flask shaker at the density of 0.5 × 10^6^ cells/mL. Cell counting was conducted daily to determine the cell density and viability. When the cell viability closed to 50%, the conditioned media were harvested for purification process. Cell culture supernatant was centrifuged and followed by filtration. The recombinant HPS protein was captured from the cell culture supernatant by 5 mL of Q HP column (GE Healthcare, Uppsala, Sweden) and eluted by 130 mM of NaCl in 20 mM of Tris-HCl (pH7.0) buffer. The eluted HPS protein was dialyzed into PBS and stored at −80 °C. 3 batches of recombinant HPS protein were analyzed to confirm its quality: purity was analyzed by SDS-PAGE and RP-HPLC (Agilent 1260, St. Clara, CA, USA); N-terminal amino acid sequence analysis were performed through Edman degradation method using Edman Sequencer PPSQ-31B (SHIMAZU, JP); sequence coverage were analyzed through peptide mass fingerprinting using Q Exactive HF (Thermo Fisher Scientific, Waltham, MA, USA) and nanoLC UltiMate3000 (Dionex, Sunnyvale, CA, USA) by National Engineering Research Center for Protein Drug (Beijing, China); HCP was analyzed by enzyme-linked immunosorbent assay (ELISA) using third-generation ELISA (F550, Cygnus Technologies, Southport, NC, USA); HCD was analyzed by Q-PCR using DNA extraction and amplification kit (D555T, Cygnus Technologies, Southport, NC, USA); endotoxin detection was performed by colorimetric method using LAL (TAL) reagent (Xiamen Bioendo Technology, Xiamen, China).

### 4.2. Immunoblotting

Immunoblotting for proteins was performed by standard methods, using the following antibodies: HPS (1:500, Invitrogen, PA5-30030), ERK1/2 (C9) (1:200, Santa Cruz, sc-514302), *p*-ERK (E4) (1:200, Santa Cruz, sc-7383).

### 4.3. In Vitro Hepatocytes Proliferation Assay

Mouse primary hepatocytes were isolated using a standard 2-step collagenase perfusion as previous described [24], and cultured in Hepatozyme serum-free media (GIBCO, 17705021) according to the manufacturer’s instruction. Human hepatic cell line WRL68 were purchased from the American Type Culture Collection (ATCC), and cultured in RPMI 1640 medium. After 12-h starvation, cells were treated with HPS for 24h. Hepatocytes proliferation was determined by MTT assay (Abcam, ab211091).

### 4.4. Mice and Treatment

Male BALB/c wild-type (WT) mice and C57BL/6J WT mice (8 weeks old, body weight 20 ± 2.0g) were purchased from Beijing Vital River Laboratory Animal Technology. Male ob/ob mice in C57BL/6J background were purchased from Nanjing Biomedical research institute of Nanjing University. All mice were kept at the Animal Facility of our institution under specific pathogen-free conditions and had access to food and water ad libitum and the experiments were reviewed and approved by the Institutional Animal Care and Use Committee of our institution (IACUC2020598, approved: 1 March 2020).

The optimal dose of CCl_4_ or D-GalN/LPS for induction of lethal or non-lethal hepatotoxicity was identified as previous described methodology [5,25]. For CCl_4_-induced non-lethal hepatotoxicity model, mice were injected intraperitoneally with 0.1 mL of CCl_4_ in corn oil (1:100 *v*/*v*) per 1 kg of body weight. For the D-GalN/LPS model, mice were induced by intraperitoneal injections of 500 mg/kg D-GalN (SigmaSt Louis, MO, USA, G0500), followed by intraperitoneal injections of 20µg/kg LPS (Sigma, L6529, St Louis, MO, USA). For survival analysis, a lethal dose of 2.5 mL/kg if CCl_4_ or 1000 mg/kg of D-GalN plus 50µg/kg of LPS was injected intraperitoneally. Treatment with HPS was performed by intraperitoneal injections HPS 12 h before and 24 and 48 h after CCl_4_ injection or 12 h before D-GalN/LPS injection with indicated dose (0.2, 1.0, 5.0mg/kg, respectively). The dose of HPS was chosen based on our previous reports [26].

### 4.5. HPS Treatment of Monkeys with Acute Liver Injury

Male cynomolgus monkeys aged 4–6 years (3–4 kg) were purchased from Kunming Yaling Biotechnology Co., Ltd. (Kunming, China). Standard laboratory chow and water were given ad libitum. All animals were housed in singular standard cages in an air-conditioned room (21–25 °C), with a 12-h light/dark cycle. The monkey experiments were reviewed and approved by SHANDONG XINBO drug analysis and testing center (XB-IACUC-2018-0020). The health status of each monkey was determined by the local veterinary department. The non-lethal hepatotoxicity was induced in the rhesus monkeys with intraperitoneal administration of D-GalN (Sigma–Aldrich, St Louis, MO, USA) as described elsewhere [20]. Animals were then allowed to move and eat freely in cages. Briefly, ten cynomolgus monkeys were randomly assigned to the experimental group or control group. The D-GalN solution was injected intravenously into the experimental monkeys within 10 min at a dose of 300 mg/kg and 0.6 mL/kg. For HPS treatment, 12 h before and 24, 48, and 72 h after D-GalN injection, HPS was injected intravenously into the monkeys, the dose was 0.25 mg/kg, and the volume was 2.5 mL/kg, while the control group received a PBS injection. The efficacious dose in monkey was estimated indirectly based on mice studies. At 96 h after D-GalN injection, serum was collected to detect ALT, AST, TBIL, DBIL, and ALP. Liver biopsy was performed on each group of animals for pathological examination.

### 4.6. Histological Analysis

Liver tissues were fixed in 10% paraformaldehyde and embedded in paraffin. Sections were stained with hematoxylin and eosin (H&E) using standard procedures. The necrosis was expressed as a percentage of necrotic areas of ×200 magnification per slide [25]. Hepatocytes’ proliferation were analyzed by PCNA staining (anti-PCNA antibody, ab46545, 1:100). PCNA-positive cells were analyzed from randomly selected 5 fields of ×200 magnification for each sample. Hepatocyte apoptosis analysis were performed using TUNEL assay kit (Roche, 11684817910) according to manufactory’s instruction. TUNEL-positive cells were analyzed from randomly selected 5 fields of ×200 magnification for each sample. Hepatic ROS levels in mice were assayed by DHE staining (GPI18243, Beijing genepool Biotechnology Co., LTD, China) in accordance with the manufacturer’s protocol. Hepatic lipid peroxidation in monkey were analyzed by 4-HNE staining (ab48506, 1:50 dilution). 4-HNE-positive cells were analyzed from randomly selected 5 fields of ×200 magnification for each sample. All images of the liver sections were captured using a Nikon Digital Sight DS-U3 camera. Image analysis procedures were performed with Image J 1.49m (National Institutes of Health).

### 4.7. Analysis of Mouse Tissue Samples

For tissue chemistry analysis, snap-frozen mouse liver (100 mg) was homogenized in 0.5 mL of ice-cold PBS containing protease inhibitors (Complete, Roche) using high-throughput tissue grinder. The tissue homogenates were centrifuged at 12,000 rpm at 4 °C for 10 min. Hepatic MDA was quantified using the MDA Assay Kit (Nanjing Jiancheng Bioengineering institute). Hepatic ROS was quantified using the BBoxiProbe DHE (BestBio) Kit. Hepatic protein carbonylation was quantified using the Protein Carbonyl Content Assay Kit (Abcam, ab126287).

### 4.8. Analysis of Serum

Serum ALT, AST, TBIL, DBIL, ALP, BUN, CREA, and CK levels were measured with an autoanalyzer (Dimension RxL Max automatic biochemical analyzer). The serum TNF-α, IFNγ, IL-6, and MCP1 levels were measured using a BD Cytometric Bead Array (BD Bioscience, San Diego, CA) in accordance with manufacturer’s instructions. Serum 4-HNE was measured using ELISA Kit (Abcam, ab238538).

### 4.9. RNA Isolation and Quantitative RT-PCR Analysis

RNA isolation and quantitative RT-PCR analysis for *TNF-α*, *IL-6*, *IFNγ, IL-1β, CAT*, and *SOD2* were performed as previously described (17). Gene expression was determined using the ΔΔCT calculation and mRNA levels are expressed as fold induction represented the relative expression of the target genes in HPS-treated group over that of PBS-treated group. TATA-box binding protein (*TBP*) was used as internal control. Specific primers for each gene were listed in Table 3 [27,28].

### 4.10. Pharmacokinetic Analysis

HPS was administrated into SD rats (Beijing Vital River Laboratory Animal Technology) by intravenous or subcutaneous injection (3.5 mg/kg, *n* = 3~4/group). Blood samples were collected from the venous plexus of the rat eyes at 1, 5, 15, and 30 min, and 1, 2, 4, 6, 8, 12, and 24 h after administration. Plasma was separated by centrifugation at 3000 rpm for 20 min and analyzed using HPS ELISA Kit (EH239RB, Thermo Fisher). Samples were analyzed in duplicate and when appropriate, at multiple dilutions. The standard curve range for the assays was 3–500 ng/mL.

For tissue distribution analysis, HPS was labeled with ^125^I and administrated into SD rats by subcutaneous injection (3.5 mg/kg, *n* = 5/group). Tissue samples were collected at 2, 8, 24, and 48 h after administration and rinsed thoroughly. We added an equal volume of 20% TCA to the tissue homogenate to precipitate the protein to determine the total radioactivity of each tissue. After centrifugation, the supernatant was aspirated, and the radioactivity of the portion precipitated by TCA was measured.

For urinary and fecal excretion analysis, SD rats were subcutaneously injected with ^125^I-labeled HPS (3.5 mg/kg, *n* = 6/group) and housed in metabolic cage individually. At 24, 48, 72, 96, 120, 144, 168, 192, 240, 288, 336, and 384 h after administration, urine and feces were collected, the volume was recorded, the radioactivity per unit volume was measured, and the percentage of emitted radioactivity to the injected radioactivity was calculated.

### 4.11. Acute Toxicity Study

Male BALB/c WT mice at 8 weeks of age were randomly assigned to HPS treatment or vehicle groups, and were intravenously injected with various amounts of HPS (5.0, 10.0, or 20.0 mg/kg) as indicated or vehicle one time. 7 days later, the mice were killed after a 4h fasted. Blood was collected by heart puncture to separate the serum for biochemical analysis. Blood cell distribution were analyzed with an automated blood cell counter (MEK-7222K, Nihon Kohden). Tissue samples were fixated in 10% phosphate-buffered formaldehyde.

Male C57BL/6J WT and ob/ob mice at 8 weeks of age were divided into two groups with one group receiving intraperitoneal injection of PBS and another group receiving HPS (1 mg/kg body weight) every day for 1 week. Hepatic triglyceride levels were measured following two-step extraction with chloroform methanol [29]. Liver sections were stained with H&E and scored using the nonalcoholic fatty liver disease (NAFLD) activity score system (NAS) as described previously [30].

### 4.12. Statistical Analysis

Data are expressed as mean ± SD. Survival time was analyzed using Kaplan–Meier, and significance was tested with the log-rank test. Data were compared among groups using a Student’s *t*–test. In case the data did not follow a normal distribution, a Mann–Whitney test was used. *p*-value < 0.05 was considered statistically significant.

## Figures and Tables

**Figure 1 ijms-22-12886-f001:**
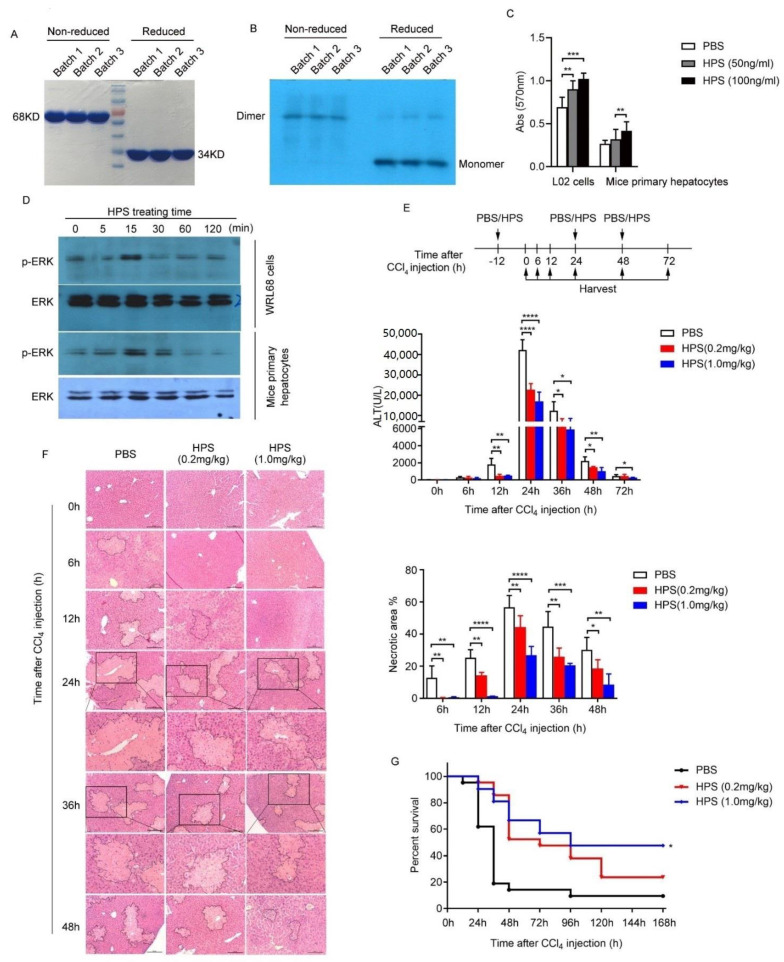
Expression and characterization of recombinant human hepassocin produced by CHO cells. (**A**) SDS-PAGE analysis of purity of 3 batches of HPS protein. The HPS molecular weight of the naturally active form (dimer form) is 68 KD and the molecular weight of the monomer is 34 KD. (**B**) Immunoblotting analysis of HPS protein using human HPS antibody. (**C**) WRL68 cells and primary hepatocytes were treated with 50 or 100 ng/mL HPS as indicated for 24 h. MTT was performed to detect cell proliferation. Data are represented the results of three independent experiments and expressed as mean ± SEM. A Student’s *t*-test was used to compare the mean relative values between groups. (** *p* < 0.01, *** *p* < 0.001). (**D**) WRL68 cells and primary mouse hepatocytes were serum starved overnight, following by treated with 100 ng/mL of HPS for indicated times. Immunoblotting was performed to detected ERK phosphorylation. Total ERK was used as control. For panel (**E**) and panel (**F**) BALB/c male mice at 8 weeks of age were intraperitoneally injected with various amounts of HPS (0.2 or 1.0 mg/kg) or PBS 12 h before and 24 and 48 h after 100 µL/kg of CCl_4_ (1:100 diluted in corn coil) injection as indicated. (**E**) Mice sera were harvested, and ALT levels were detected at indicated time points after CCl_4_ injection. (*n* = 5/group). (**F**) Liver tissues were fixed, sectioned, and stained with H&E for histopathological and morphological analysis at indicated time points after CCl_4_ injection. Scale bar, 200 µm. The part enclosed by a dashed line represents the necrotic area. Part of the necrotic area in the 24-h group and 36-h group is enlarged and displayed below. The percentage of necrotic area was quantitated using ImageJ software and values are the mean ± SD of five fields of measurements. For panel (**E**) and panel (**F**), data are represented as mean ± SD. A Student’s *t*-test was used to compare the mean values between groups. (* *p* < 0.05, ** *p* < 0.01, *** *p* < 0.001, **** *p* < 0.0001). (**G**) BALB/c male mice at 8 weeks of age were intraperitoneally injected with various amounts of HPS (0.2 or 1 mg/kg) or PBS 12 h before and 24 and 48 h after 2.5 mL/kg CCl_4_ (1:100 diluted in corn coil) injection. Cumulative survival analysis was determined with a Kaplan–Meier diagram (*n* = 10/group). Log rank test was used to compare the survival distributions between groups (**p* < 0.05).

**Figure 2 ijms-22-12886-f002:**
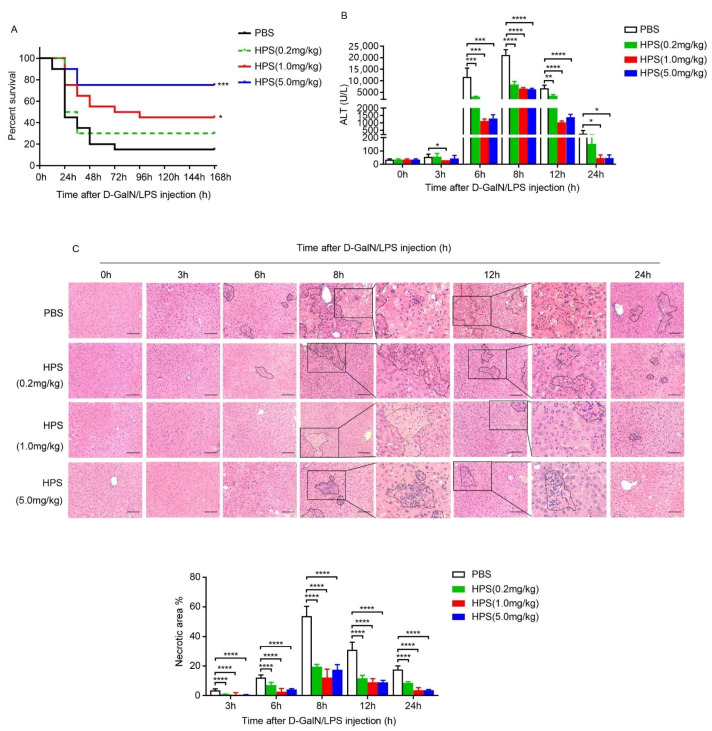
HPS treatment attenuated the damage of D-GalN/LPS-induced liver injury in mice. (**A**) BALB/c male mice at 8 weeks of age were intraperitoneally injected with various amounts of HPS (0.2, 1.0, or 5.0 mg/kg) or PBS 12 h before and 24 and 48 h after D-GalN (1000 mg/kg) and LPS (50 µg/kg) injection (*n* = 10/group). Cumulative survival analysis was determined with a Kaplan–Meier diagram. Log rank test was used to compare the survival distributions between groups (* *p* < 0.05, *** *p* < 0.001). For panel (**B**) and panel (**C**), BALB/c male mice at 8 weeks of age were intraperitoneally injected with various amounts of HPS (0.2, 1.0, or 5.0 mg/kg) or PBS 12 h before D-GalN (500 mg/kg) and LPS (20 µg/kg) injection (*n* = 5/group). (**B**) Mice serum ALT levels were detected at indicated time points after D-GalN/LPS injection. Data are represented as mean ± SD. A Student’s *t*-test was used to compare the mean values between groups. (* *p* < 0.05, ** *p* < 0.01, *** *p* < 0.001, **** *p* < 0.0001). (**C**) Liver tissues were fixed, sectioned and stained with H&E for histopathological and morphological analysis at indicated time points after D-GalN/LPS injection. Scale bar, 100 µm. The part enclosed by a dashed line represents the necrotic area. Part of the necrotic area in the 8-h group and 12-h group is enlarged and displayed on the right. The percentage of necrotic area was quantitated using ImageJ software and values are the mean ± SD of five fields of measurements. A Student’s *t*-test was used to compare the mean values between groups. (**** *p* < 0.0001).

**Figure 3 ijms-22-12886-f003:**
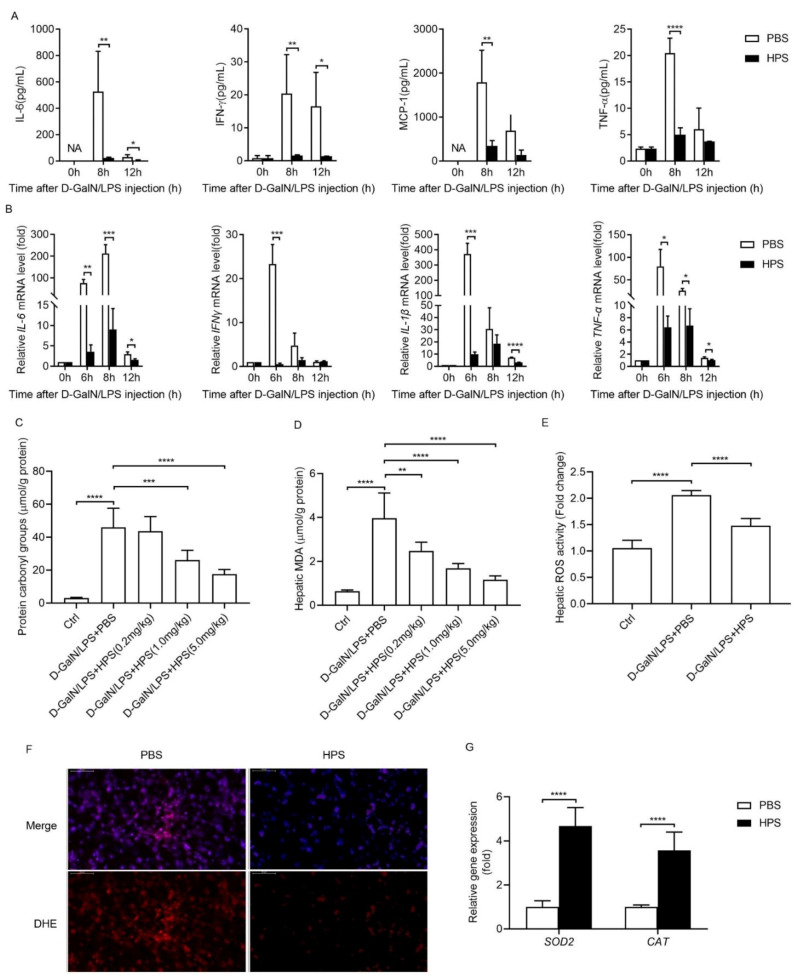
HPS treatment inhibits pro-inflammatory cytokine production and hepatic oxidative stress in mice after D-GalN/LPS injection. BALB/c male mice at 8 weeks of age were intraperitoneally injected with 500 mg/kg D-GalN, followed by intraperitoneal injections of 20 µg/kg LPS. (**A**) Serum levels of IL-6, IFNγ, MCP1, and TNFα of PBS or HPS (1.0mg/kg)-treated mice were determined at indicated time points after D-GalN/LPS injection by FACS using Mouse Inflammation Kit (BD Cytometric Bead Array (CBA)) (*n* = 5/group). (**B**) The mRNA levels of the *IL-6*, *IFNγ*, *IL-1β*, and *TNF-α* in the hepatic homogenate of PBS or HPS (1.0 mg/kg)-treated mice were determined at indicated time points after D-GalN/LPS injection by real-time PCR. The expression levels for the target genes were normalized to TBP. (**C**) Changes in protein carbonyl content in the hepatic homogenate of different dose of HPS-treated mice were detected at 6 h after D-GalN/LPS injection by colorimetric method. (**D**) Hepatic MDA levels of different dose of HPS-treated mice were detected at 6 h after D-GalN/LPS injection by colorimetric method. (**E**) Hepatic ROS levels of PBS or HPS (1.0 mg/kg)-treated mice were determined at 6 h after D-GalN/LPS injection by colorimetric method. (**F**) Representative images of immunofluorescence analysis of DHE-stained liver sections of PBS or HPS (1.0 mg/kg)-treated mice at 6 h after D-GalN/LPS injection. Scale bar, 50 µm. (**G**) The mRNA levels of the *SOD2* and *CAT* in the hepatic homogenate of PBS or HPS (1.0 mg/kg)-treated mice were determined at 6 h after D-GalN/LPS injection by real-time PCR. The expression levels for the target genes were normalized to TBP. All data are represented as mean ± SD. A Student’s *t*-test was used to compare the mean values between groups. (* *p* < 0.05, ** *p* < 0.01, *** *p* < 0.001, **** *p* < 0.0001).

**Figure 4 ijms-22-12886-f004:**
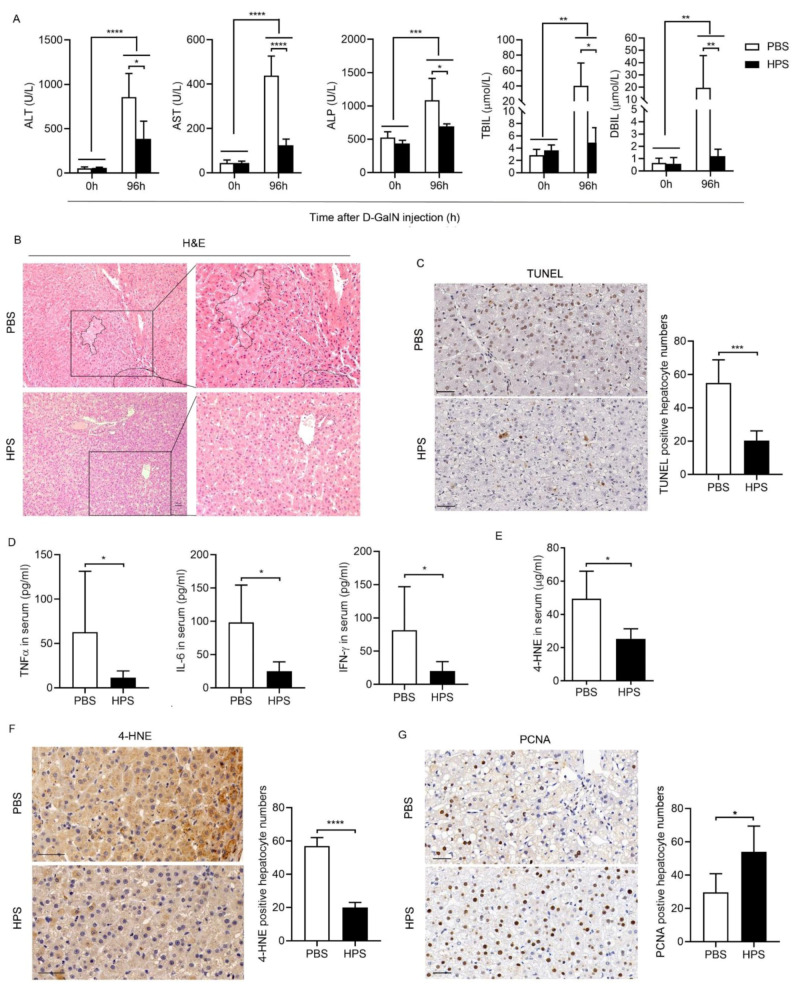
Peripheral delivery of recombinant human HPS prevents acute liver injury in in nonhuman primates. Cynomolgus monkey were injected intravenously with 300 mg/kg D-GalN. HPS (0.25 mg/kg) or PBS were injected intravenously into the monkey at 12 h before and 24, 48, 72 h after D-GalN injection (*n* = 5/group). (**A**) Serum ALT, AST, ALP, TBIL, and DBIL levels at 96 h after D-GalN injection were detected. (**B**) Representative images of H&E stained-liver sections of the monkey at 96 h after D-GalN injection. The part enclosed by a dashed line represents the necrotic area, which is enlarged and displayed on the right. Scale bar, 50 µm. (**C**) Representative images of immunohistochemistry analysis of TUNEL stained liver sections of the monkey at 96 h after D-GalN injection and the number of TUNEL positive cells of five fields of measurements. Scale bar, 50 µm. (**D**) Serum levels of IL-6, IFNγ, and TNF-α were determined at 96 h after D-GalN injection by FACS using nonhuman primate Th1/Th2 Cytokine Kit (BD Cytometric Bead Array (CBA)). (**E**) Serum levels of 4-HNE were quantified at 96 h after D-GalN injection by colorimetric detection using a competitive ELISA Kit. (**F**) Representative images of immunohistochemistry analysis of 4-HNE-stained liver sections of the monkey at 96 h after D-GalN injection and the number of 4-HNE positive cells of five fields of measurements. Scale bar, 50 µm (**G**) Representative images of immunohistochemistry analysis of PCNA stained liver sections of the monkey at 96h after D-GalN injection and the number of PCNA positive cells of five fields of measurements. Scale bar, 50µm. All data are represented as mean ± SD. For panel (**A**), panel (**D**), and panel (**E**), Mann–Whitney test was used to compare the mean relative values between groups (* *p* < 0.05). For panel (**C**), panel (**F**), and panel (**G**), Student’s *t*-test was used to compare the mean relative values between groups. (* *p* < 0.05, ** *p* < 0.01, *** *p* < 0.001, **** *p* < 0.0001).

**Figure 5 ijms-22-12886-f005:**
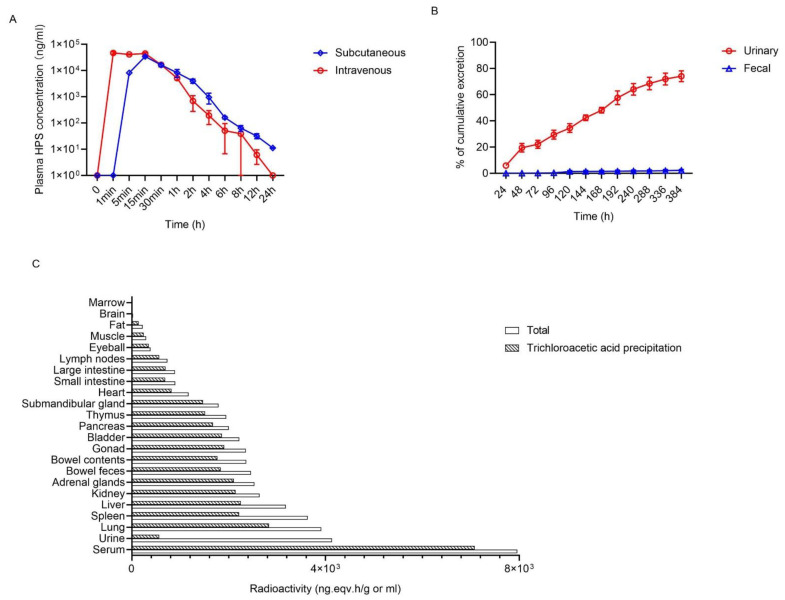
The pharmacokinetics of HPS. (**A**) Plasma concentration-time profiles after intravenous or subcutaneous administration of HPS (3.5 mg/kg) in rats (*n* = 3/group). Data are presented as the mean ± SD. (**B**) Tissue distribution of HPS after intravenous administration in rats (*n* = 5/group). (**C**) Cumulative urinary excretion and fecal elimination as a percent of dose administrated (*n* = 6/group).

**Figure 6 ijms-22-12886-f006:**
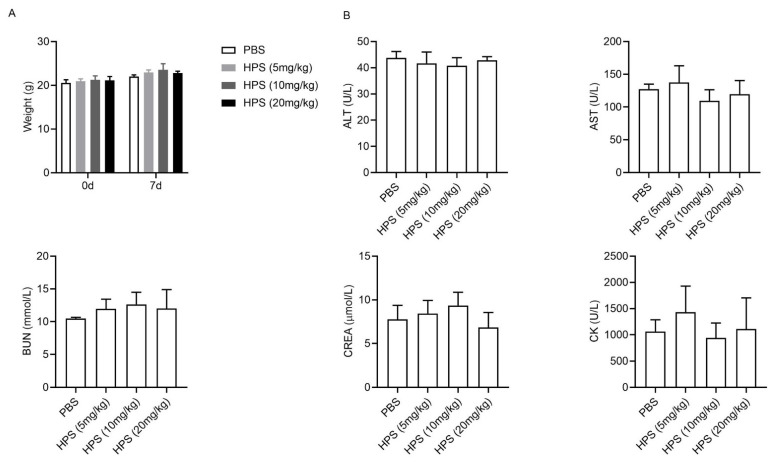
Effects of a single dose of HPS on tissue injury in mice. BALB/c male mice at 8 weeks of age were intravenously injected with various amounts of HPS (5.0, 10.0, or 20.0 mg/kg) or PBS (*n* = 5/group). Seven days later, primary safety evaluation was analyzed. (**A**) Body weight. (**B**) Serum ALT, AST, BUN, CREA, and CK levels. (**C**) Routine blood tests. (**D**) Liver tissues were fixed, sectioned and stained with H&E for histopathological and morphological analysis. Scale bar, 200 µm.

**Figure 7 ijms-22-12886-f007:**
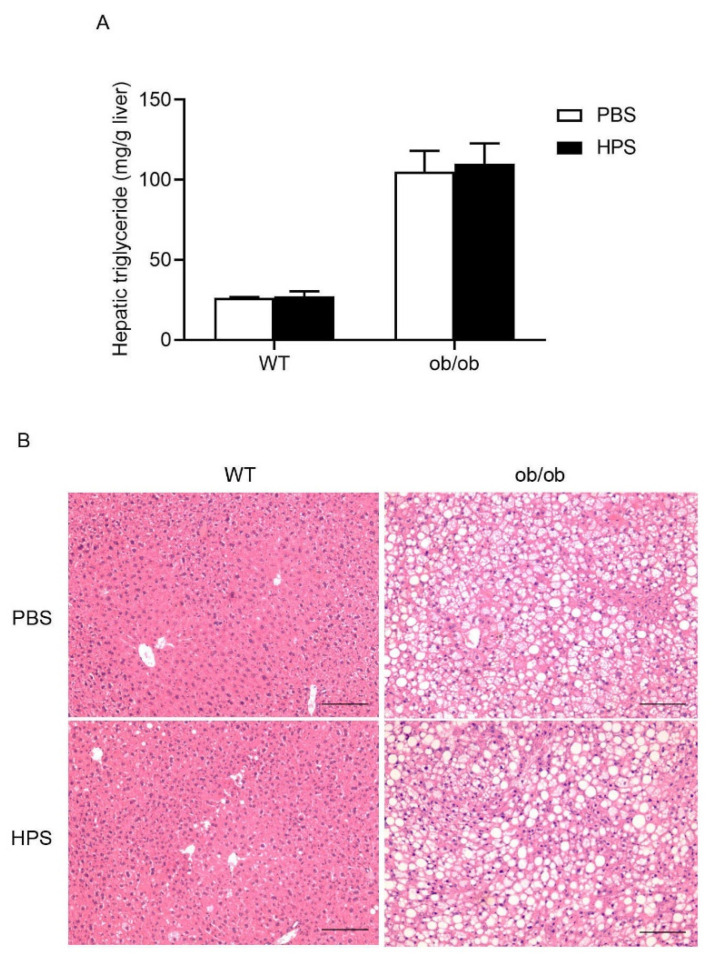
Effects of administration of HPS in lipid accumulation in wild type and ob/ob mice. Male C57BL/6J WT and ob/ob mice at 8 weeks of age were divided into two groups with one group receiving intraperitoneal injection of PBS and another group receiving HPS (1mg/kg body weight) every day for 1 week (*n* = 5/group). (**A**) Hepatic triglyceride levels were determined. (**B**) Liver tissues were fixed, sectioned and stained with H&E for histopathological and morphological analysis. Scale bar, 200 µm. (**C**) NAFLD activity score (steatosis, ballooning degeneration, and inflammation).

**Table 1 ijms-22-12886-t001:** Critical quality attributes of HPS.

Test Items	Results
Purity	95.29 ± 0.45%
N-terminal amino acid sequence	LED[C]AQEQMRLRAQV
HCP	153 ± 8.39 ng/mg HPS protein
HCD	0.8 ± 0.04 ng/mg HPS protein
Endotoxin	<0.2 EU/mg HPS protein

HCP, host cell protein; HCD, host cell DNA.

**Table 2 ijms-22-12886-t002:** Summary of pharmacokinetic parameters for HPS in mice.

	Intravenous	Subcutaneous
Dose (mg/kg)	3.5	3.5
AUC (µg/h/mL)	27.79 ± 2.66	28.96 ± 3.15
CL (ml/h/kg)	135.94 ± 14.70	127.25 ± 18.63
T1/2 (h)	1.57	1.51
Cmax (µg/mL)	47.78 ± 5.01	33.93 ± 2.06
Tmax (h)	0.0167	0.25
%F	NA	100

AUC, area under the plasma concentration curve; CL, clearance; T1/2, half-life; Cmax, maximal plasma concentration; Tmax, time to Cmax; F%, absolute bioavailability; NA, not available.

**Table 3 ijms-22-12886-t003:** Sequences of primers used for quantitative PCR.

Gene Name	Forward (5′-3′)	Reverse (5′-3′)
*CAT*	AGCGACCAGATGAAGCAGTG	TCCGCTCTCTGTCAAAGTGTG
*IFNγ*	ATGAACGCTACACACTGCATC	CCATCCTTTTGCCAGTTCCTC
*IL-1β*	GCAACTGTTCCTGAACTCAACT	ATCTTTTGGGGTCCGTCAACT
*IL-6*	CTGCAAGAGACTTCCATCCAG	AGTGGTATAGACAGGTCTGTTGG
*SOD2*	CAGACCTGCCTTACGACTATGG	CTCGGTGGCGTTGAGATTGTT
*TBP*	AGAACAATCCAGACTAGCAGCA	GGGAACTTCACATCACAGCTC
*TNF-α*	CTGAACTTCGGGGTGATCGG	GGCTTGTCACTCGAATTTTGAGA

## Data Availability

Data are contained within the article.

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
