# Peer review of "Recombinant Human HPS Protects Mice and Nonhuman Primates from Acute Liver Injury"

_ijms, 2021, doi:10.3390/ijms222312886_

Round 1
Reviewer 1 Report
This study demonstrated that “Recombinant human HPS protects mice and nonhuman primates from acute liver injury.” This manuscript is interesting, however, there are several concerns relating that should be carefully address by the authors.
Results
2.1. Expression and characterization of recombinant human HPS produced by CHO cells
2.2. HPS treatment attenuates D-galactosamine plus lipopolysaccharide(D-GalN/LPS)-induced liver injury in mice
→ Figure 1E and 2C indicated the necrosis areal density. How calculated the necrosis areal density?
Authors should describe about it in these sections or Materials and Methods.
Materials and Methods
4.4. Mice and treatment
4.5. HPS treatment of monkeys with acute liver injury
→ Why were different the method for acute liver injury (injection, concentration etc.) using between mice and monkeys? Authors should describe about it in more detail.
Abstract
Authors described that “Our results suggest that exogenous HPS has protective effects on acute liver injury in both mice and monkeys except rats.”
→ Why these phenomena were occurred?
If possible, authors should discuss about it in Discussion.
Author Response
Dear reviewer:
We are extremely grateful to the reviewers’ serious and constructive comments on our manuscript. We have carefully read the thoughtful comments and found that these suggestions are helpful for us to improve our manuscript. Based on the enlightening questions and helpful advices, we have now completed the revision of our manuscript. The point to point responds to the reviewers’ comments are listed as following:
Point 1. authors should describe about how calculated the necrosis areal density in Figure legend or Materials and Methods.
Response: Thanks for the valuable and thoughtful comments. As suggested by the reviewer, we describe about how calculated the necrosis areal density in Materials and Methods in detail and the relevant reference is supplemented in the revised manuscript.
Page 18:
Liver tissues were fixed in 10% paraformaldehyde and embedded in paraffin. Sections were stained with hematoxylin and eosin (H&E) using standard procedures. The necrosis was expressed as a percentage of necrotic areas of ×200 magnification per slide [25]. Hepatocytes proliferation was analyzed by PCNA staining (anti-PCNA antibody, ab46545, 1:100). PCNA-positive cells were analyzed from randomly selected 5 fields of ×200 magnification for each sample. Hepatocyte apoptosis analysis were performed using TUNEL assay kit (Roche, 11684817910) according to manufactory’s instruction. TUNEL-positive cells were analyzed from randomly selected 5 fields of ×200 magnification for each sample. Hepatic ROS levels in mice were assayed by DHE staining (GPI18243, Beijing genepool Biotechnology Co., LTD, China) according to the manufacturer’s protocol. Hepatic lipid peroxidation in monkeys were analyzed by 4-HNE staining (ab48506, 1:50 dilution). 4-HNE-positive cells were analyzed from randomly selected 5 fields of ×200 magnification for each sample. All images of the liver sections were captured using a Nikon Digital Sight DS-U3 camera. Image analysis procedures were performed with Image J 1.49m (National Institutes of Health).
Point 2. Why were different the method for acute liver injury (injection, concentration etc.) using between mice and monkeys? Authors should describe about it in more detail.
Response: As suggested by the reviewer, we describe the process of animal experiment in more detail and supplement the relevant references. The optimal dose of CCl4 or D-GalN/LPS for induction of lethal or non-lethal hepatotoxicity was identified based on a series of pre-experiments using methodology described in our previous paper (1). The non-lethal hepatotoxicity model of monkeys was induced by intraperitoneal administration of D-GalN as previous described (2). The efficacious dose of HPS in mice was chosen based on our previous reports (3). The dose of HPS in monkey was estimated indirectly based on mice studies.
Injection of substances into the peritoneal cavity is a common technique in laboratory rodents but rarely is used in larger mammals and humans. Intraperitoneal delivery is considered a parenteral route of administration, its primary route of absorption is into the mesenteric vessels, which drain into the portal vein and pass through the liver. Because of its simple operation, intraperitoneal injection is often used in small animal experiments, whereas it is related to severe complications such as abdominal adhesions, gastrointestinal injury, and high mortality rate in monkeys according to previous reports (4). Therefore, in monkey experiments, we used intravenous administration. Importantly, intravenous administration of HPS is more suitable for clinical use in human.
Page 18:
The optimal dose of CCl4 or D-GalN/LPS for induction of lethal or non-lethal hepatotoxicity was identified as previous described methodology [5, 25]. For CCl4 induced non-lethal hepatotoxicity model, mice were injected intraperitoneally with 0.1ml of CCl4 in corn oil (1:100 v/v) per 1 kg of body weight. For D-GalN/LPS model, mice were induced by intraperitoneal injections of 500mg/kg D-GalN (Sigma, G0500), followed by intraperitoneal injections of 20μg/kg LPS (Sigma, L6529). For survival analysis, a lethal dose of 2.5ml/kg CCl4 or 1000mg/kg D-GalN plus 50μg/kg LPS was injected intraperitoneally. Treatment with HPS was performed by intraperitoneal injections HPS 12h before and 24h and 48h after CCl4 injection or 12h before D-GalN/LPS injection with indicated dose (0.2mg/kg, 1.0mg/kg, 5.0mg/kg, respectively). The dose of HPS was chosen based on our previous reports [26].
Page 18:
Male cynomolgus monkeys aged 4-6 years (3-4kg) were purchased from Kunming Yaling Biotechnology Co., Ltd. (Kunming, China). Standard laboratory chow and water were given ad libitum. All animals were housed in singular standard cages in an air-conditioned room (21-25℃), with a 12h light/dark cycle. The monkey experiments were reviewed and approved by SHANDONG XINBO drug analysis and testing center (XB-IACUC-2018-0020). The health status of each monkey was determined by the local veterinary department.
The non-lethal hepatotoxicity was induced in the rhesus monkeys with intraperitoneal administration of D-GalN (Sigma-Aldrich, CA, USA) as described elsewhere [20]. Animals were then allowed to move and eat freely in cages. Briefly, ten cynomolgus monkeys were randomly assigned to the experimental group or control group. The D-GalN solution was injected intravenously into the experimental monkeys within 10 minutes at a dose of 300mg/kg and 0.6 ml/kg. For HPS treatment, 12h before and 24h, 48h and 72h after D-GalN injection, HPS was injected intravenously into the monkeys, the dose was 0.25 mg/kg, and the volume was 2.5 ml/kg, while the control group received a PBS injection. The efficacious dose in monkey was estimated indirectly based on mice studies. At 96h after D-GalN injection, serum was collected to detect ALT, AST, TBIL, DBIL, and ALP. Liver biopsy was performed on each group of animals for pathological examination.
Reference:
(1) Li, C. Y.; Cao, C. Z.; Xu, W. X.; Cao, M. M.; Yang, F.; Dong, L.; Yu, M.; Zhan, Y. Q.; Gao, Y. B.; Li, W.; Wang, Z. D.; Ge, C. H.; Wang, Q. M.; Peng, R. Y.; Yang, X. M., Recombinant human hepassocin stimulates proliferation of hepatocytes in vivo and improves survival in rats with fulminant hepatic failure. Gut 2010, 59, (6), 817-826.
(2) Zhang, W.; Tao, S. S.; Wang, T.; Li, Y. T.; Chen, H.; Zhan, Y. Q.; Yu, M.; Ge, C. H.; Li, C. Y.; Ren, G. M.; Yin, R. H.; Yang, X. M., NLRP3 is dispensable for d-galactosamine/lipopolysaccharide-induced acute liver failure. Biochemical and biophysical research communications 2020, 533, (4), 1184-1190.
(3) Gao, M.; Zhan, Y. Q.; Yu, M.; Ge, C. H.; Li, C. Y.; Zhang, J. H.; Wang, X. H.; Ge, Z. Q.; Yang, X. M., Hepassocin activates the EGFR/ERK cascade and induces proliferation of L02 cells through the Src-dependent pathway. Cellular signalling 2014, 26, (10), 2161-6
(4) Turner, P. V.; Brabb, T.; Pekow, C.; Vasbinder, M. A., Administration of substances to laboratory animals: routes of administration and factors to consider. Journal of the American Association for Laboratory Animal Science: JAALAS 2011, 50, (5), 600-13.
Point 3. Why authors described that “Our results suggest that exogenous HPS has protective effects on acute liver injury in both mice and monkeys except rats.” in abstract.
Response: We are sorry for this mistake. Since we did not describe it clearly, it caused a misunderstanding to the reviewers. In the manuscript, we evaluated the effects of HPS administration on the pathogenesis of acute liver injury in monkey and mice. What we want to describe is: In addition to previous reports showing that HPS has a protective effect on acute liver injury in rats, our results also suggest that exogenous HPS has a protective effect on acute liver injury in mice and monkeys. In the revised manuscript, we make changes.
Page 1:
Our results suggest that exogenous HPS has protective effects on acute liver injury in both mice and monkeys.
Reviewer 2 Report
The present study describes protective roles of Hepassocin (HPS) in acute liver failure.
The authors create recombinant human HPS in CHO cells and purify it using chromatography techniques. They achieved a 95% purity. They characterize the purified HPS in vitro and in vivo and observe protective effects in mice against CCl4 induced liver damage. The authors observed significantly higher mice survival with 1.0mg/kg dose of HPS.
The authors further characterize protective effects of HPS in D-GalN/LPS induced acute liver injury model in mice. The authors find significant higher survival in mice and significant reduction in ALT levels upon HPS administration. The authors find less necrosis in livers of HPS treated mice. Transcriptomic and serum levels of IL-6, IFNγ, MCP-1 and TNF-α were also found to be significantly downregulated for most of the time points.
rhHPS also showed protective effect in D-GalN/LPS induced acute liver injury in nonhuman primates. Levels of ALT, AST, ALP etc were found to be decreased in serum along with IL-6, IFNγ and TNF-α and 4-HNE. The authors show less cell death and higher proliferation of hepatocytes.
Plasma concentration time profiles, excretion and tissue distribution were analyzed for intravenous and subcutaneous administration in rats.
The authors further investigate for acute toxicity and do not find any detrimental effects.
Overall, an interesting study indicating protective effects of HPS in liver injury.
I have a few minor questions and suggestions.
- Hepassocin is known to cause increase in hepatic lipid accumulation and promotes NAFLD. Do the authors have any data supporting otherwise? Or do the authors see changes in NAS scores upon long term administration of rhHPS in high doses as suggested for therapeutic effects? The authors should include data supporting that sustained high levels of rhHPS do not show changes in NAS scores or progress to NAFLD.
- The mechanism how HPS is increasing cell proliferation and decreasing cell death is not clear from this study. Is the mechanism known? Have the authors examined the levels of reactive oxygen species in the liver tissue sections? DCFDA staining should be performed on liver tissue sections treated with PBS or HPS. Is HPS known to regulate any ROS scavenging genes such as MnSOD or Catalase? Gene expression analysis should be performed for the major ROS scavengers.
- The authors claim lesser necrosis in HPS treated liver sections in mice and nonhuman primates, however, it is not apparent from the images presented, please provide better images for 1E, 2C and 4B, also include some images at higher magnification. The imageJ script that was used to quantify necrosis from H&E sections must be included in the Methods section. Alternatively, the authors may use a marker for necrosis (such as RIPK3) and perform immunohistochemistry.
- Figure legend for Figure 2 should be revised:
Line 197: (E) Mice serum ALT levels were detected at indicated time points after CCl4 injection. (n=5/group). (F) Liver tissues were fixed, sectioned and stained with H&E for histopatho- logical and morphological analysis at indicated time points after CCl4 injection. Scare bar, 50μm.
(E) and (F) should be interchanged as the figure does not match the figure legend.
- Line 195 : ‘phosphoralation’ should be changed to phosphorylation. Please correct other typos in manuscript as well.
- In general the manuscript needs minor work on grammar and spelling errors.
Author Response
Dear reviewer:
We are extremely grateful to the reviewers’ serious and constructive comments on our manuscript. We have carefully read the thoughtful comments and found that these suggestions are helpful for us to improve our manuscript. Based on the enlightening questions and helpful advices, we have now completed the revision of our manuscript. The point to point responds to the reviewers’ comments are listed as following:
Point 1. Hepassocin is known to cause increase in hepatic lipid accumulation and promotes NAFLD. Do the authors have any data supporting otherwise? Or do the authors see changes in NAS scores upon long term administration of rhHPS in high doses as suggested for therapeutic effects? The authors should include data supporting that sustained high levels of rhHPS do not show changes in NAS scores or progress to NAFLD.
Response: Thanks for the valuable and thoughtful comments. As suggested by the reviewer, we determine the effect of HPS on hepatic lipid accumulation in WT mice and steatohepatitis in ob/ob mice. The results show no increase in hepatic lipid accumulation in wild type and no effect on progress of NASH in ob/ob mice after 1 week of HPS (1mg/kg body weight) treatment, which suggest that at least short-term administration of HPS does not affect liver fat accumulation and the progression of NASH. The result is supplemented in the revised manuscript. Although previous studies showed that HPS appears to promote the progression of fatty liver disease. Some results are contradictory. The HPS gene knockout mice were reported with a global metabolic disorder phenotype including obesity and glucose tolerance defects on a standard chow diet (1), which are considered as major risk factors for non-alcoholic fatty liver diseases (NAFLD) (2). Hepatic lipid accumulation was significantly increased in HPS-deficient mice after hepatectomy, suggesting that HPS is required for metabolic stress in liver. Strikingly, HPS was shown to play a role in protecting against multiple types of liver injury in vivo or in vitro, including metabolic glucotoxicity- and lipotoxicity-induced liver damage (3-5). Thus, the roles of HPS in steatohepatitis have not been fully clarified.
Page 5:
Previous studies have reported that overexpression of HPS increases lipid accumulation in liver. However, we observed no increase in hepatic lipid accumulation in wild type (WT) and ob/ob mice after 1 week of HPS (1mg/kg body weight) treatment (Figure 7A & 7B).
Page 15:
Figure 7 Effects of administration of HPS in lipid accumulation in wild type and ob/ob mice. Male C57BL/6J WT and ob/ob mice at 8 weeks of age were divided into two groups with one group receiving intraperitoneal injection of PBS and another group receiving HPS (1mg/kg body weight) every day for 1 week (n=5/group). (A) Hepatic triglyceride levels were determined. (B) Liver tissues were fixed, sectioned and stained with H&E for histopathological and morphological analysis. Scale bar, 200μm. (C) NAFLD activity score (steatosis, ballooning degeneration and inflammation).
Page 20:
Male C57BL/6J WT and ob/ob mice at 8 weeks of age were divided into two groups with one group receiving intraperitoneal injection of PBS and another group receiving HPS (1mg/kg body weight) every day for 1 week. Hepatic triglyceride levels were measured following two-step extraction with chloroformmenthanol [29]. Liver sections were stained with H&E and scored using the nonalcoholic fatty liver disease (NAFLD) activity score system (NAS) as described previously [30].
Reference:
(1) Demchev, V.; Malana, G.; Vangala, D.; Stoll, J.; Desai, A.; Kang, H. W.; Li, Y.; Nayeb-Hashemi, H.; Niepel, M.; Cohen, D. E.; Ukomadu, C., Targeted deletion of fibrinogen like protein 1 reveals a novel role in energy substrate utilization. PloS one 2013, 8, (3), e58084.
(2) Anstee, Q. M.; Targher, G.; Day, C. P., Progression of NAFLD to diabetes mellitus, cardiovascular disease or cirrhosis. Nature reviews. Gastroenterology & hepatology 2013, 10, (6), 330-44.
(3) Li, C. Y.; Cao, C. Z.; Xu, W. X.; Cao, M. M.; Yang, F.; Dong, L.; Yu, M.; Zhan, Y. Q.; Gao, Y. B.; Li, W.; Wang, Z. D.; Ge, C. H.; Wang, Q. M.; Peng, R. Y.; Yang, X. M., Recombinant human hepassocin stimulates proliferation of hepatocytes in vivo and improves survival in rats with fulminant hepatic failure. Gut 2010, 59, (6), 817-26.
(4) Cheng, K. P.; Ou, H. Y.; Hung, H. C.; Li, C. H.; Fan, K. C.; Wu, J. S.; Wu, H. T.; Chang, C. J., Unsaturated Fatty Acids Increase the Expression of Hepassocin through a Signal Transducer and Activator of Transcription 3-Dependent Pathway in HepG2 Cells. Lipids 2018, 53, (9), 863-869.
(5) Ou, H. Y.; Wu, H. T.; Lin, C. H.; Du, Y. F.; Hu, C. Y.; Hung, H. C.; Wu, P.; Li, H. Y.; Wang, S. H.; Chang, C. J., The Hepatic Protection Effects of Hepassocin in Hyperglycemic Crisis. The Journal of clinical endocrinology and metabolism 2017, 102, (7), 2407-2415.
Point 2. The mechanism how HPS is increasing cell proliferation and decreasing cell death is not clear from this study. Is the mechanism known? Have the authors examined the levels of reactive oxygen species in the liver tissue sections? DCFDA staining should be performed on liver tissue sections treated with PBS or HPS. Is HPS known to regulate any ROS scavenging genes such as MnSOD or Catalase? Gene expression analysis should be performed for the major ROS scavengers.
Response: As suggested by the reviewer, we examined whether HPS affects D-GalN/LPS-induced oxidative stress by estimating protein carbonylation and malondialdehyde (MDA), which represents oxidative protein damage and lipid peroxidation. D-GalN/LPS significantly induced the carbonylation of hepatic proteins in mice, while HPS treatment inhibited the carbonylation of hepatic proteins in a dose-dependent manner. Injection of D-GalN/LPS induced markedly the accumulation of MDA in liver tissue of mice, while HPS treatment decreased the accumulation of MDA in a dose-dependent manner. Moreover, HPS treatment significantly reduced hepatic reactive oxygen species (ROS) levels and up-regulated superoxide dismutase 2 (SOD2) and Catalase (CAT) mRNA expressions. These results suggest that HPS treatment attenuates D-galN/LPS induced hepatic oxidative stress in mice. Furthermore, the levels of serum and hepatic 4-HNE (a marker of lipid peroxidation) positive cells were decreased in HPS-treated animals, indicating that HPS treatment reduced D-GalN induced hepatic oxidative stress in monkeys. Together, we provide new evidence to support that HPS treatment inhibits hepatic oxidative stress in acute liver injury animal. Theses results are supplemented in the revised manuscript.
Previous studies have shown that HPS is a mitogen for hepatocytes in a mitogen-activated protein kinase (MAPK)-dependent manner. Moreover, HPS has the ability of anti-apoptosis by inhibiting the upregulation of toxin-induced proapoptotic factors (BCL2-associated X protein (BAX), cleaved caspase 9, and increasing the expression level of anti-apoptotic factors (B cell leukemia/lymphoma 2 (BCL-2), BCL2-like 1 (BCL-XL)). In this study, we confirm these findings. In addition, we here found that HPS treatment inhibits hepatic oxidative stress and inflammatory factors expression in acute liver injury animal. We thus conclude that HPS improve liver injury by stimulating hepatocyte regeneration, attenuating hepatocytes apoptosis, and reducing oxidative stress response and inflammation. In revised manuscript, we rewrite the discussion to summarize the mechanism of HPS against liver injury.
Page 4:
Acute hepatitis induced by D-GalN/LPS is partly mediated by oxidative stress [19]. Therefore, we examined whether HPS affects D-GalN/LPS-induced oxidative stress by estimating protein carbonylation and malondialdehyde (MDA), which represent oxidative protein damage and lipid peroxidation. D-GalN/LPS significantly induced the carbonylation of hepatic proteins in mice, while HPS treatment inhibited the carbonylation of hepatic proteins in a dose-dependent manner (Figure 3C). HPS treatment also decreased the D-GalN/LPS induced accumulation of MDA in liver tissue of mice in a dose-dependent manner (Figure 3D). Moreover, HPS treatment significantly reduced hepatic reactive oxygen species (ROS) levels and dihydroethidium (DHE)-positive staining and up-regulated superoxide dismutase 2 (SOD2) and Catalase (CAT) mRNA expressions (Figure 3E-3G). These results suggest that HPS attenuates D-GalN/LPS-induced hepatic oxidative stress in mice.
Page 4:
The levels of serum and hepatic 4-hydroxynonenal (4-HNE) positive cells were decreased in HPS-treated animals, indicating that HPS treatment reduced D-GalN- induced hepatic oxidative stress in monkeys (Figure 4E & 4F).
Page 16:
Importantly, we found that D-GalN/LPS-induced hepatic oxidative stress was significantly attenuated by HPS administration, as evidenced by reduced the levels of ROS, proteins carbonylation and MDA accumulation, suggesting that HPS ameliorates acute hepatitis induced by D-GalN/LPS at least in partial by inhibiting oxidative stress. We further demonstrated that administration of HPS significantly alleviated D-GalN-induced acute liver injury in nonhuman primates, as evidenced by reduced serum ALT activity attenuated hepatocytes apoptosis, increased hepatocytes proliferation, and improved liver histology. In addition, HPS treatment also inhibited hepatic oxidative stress in D-GalN-treated nonhuman primates. These findings agree with previous reports claiming an important role of HPS in rats liver injury models [5], and highlight HPS delivery may be a potential therapeutic strategy for the treatment of acute liver injury in humans, mostly in patients who would need long-term treatment with potentially hepatotoxic drugs. Combined with previous reports, our results suggest that the improvement of liver injury by HPS is mainly relevant from four perspectives: firstly, by stimulating hepatocyte regeneration through activation of MAPK/ERK pathway [4]; secondly, by attenuating hepatocytes apoptosis through decreased proapoptotic protein Bax and elevated antiapoptotic protein Bcl-2 [5, 15]; thirdly, by reducing oxidative stress response; fourthly, by inhibiting inflammation.
Point 3. The authors claim lesser necrosis in HPS treated liver sections in mice and nonhuman primates, however, it is not apparent from the images presented, please provide better images for 1E, 2C and 4B, also include some images at higher magnification. The imageJ script that was used to quantify necrosis from H&E sections must be included in the Methods section. Alternatively, the authors may use a marker for necrosis (such as RIPK3) and perform immunohistochemistry.
Response: As suggested by the reviewer, we provide images for Fig.1E, 2C and 4B at higher magnification, and describe about how calculated the necrosis areal density in Materials and Methods in detail and supplement the relevant reference in the revised manuscript.
Page 18:
Liver tissues were fixed in 10% paraformaldehyde and embedded in paraffin. Sections were stained with hematoxylin and eosin (H&E) using standard procedures. The necrosis was expressed as a percentage of necrotic areas of ×200 magnification per slide [25]. Hepatocytes proliferation was analyzed by PCNA staining (anti-PCNA antibody, ab46545, 1:100). PCNA-positive cells were analyzed from randomly selected 5 fields of ×200 magnification for each sample. Hepatocyte apoptosis analysis were performed using TUNEL assay kit (Roche, 11684817910) according to manufactory’s instruction. TUNEL-positive cells were analyzed from randomly selected 5 fields of ×200 magnification for each sample. Hepatic ROS levels in mice were assayed by DHE staining (GPI18243, Beijing genepool Biotechnology Co., LTD, China) according to the manufacturer’s protocol. Hepatic lipid peroxidation in monkeys were analyzed by 4-HNE staining (ab48506, 1:50 dilution). 4-HNE-positive cells were analyzed from randomly selected 5 fields of ×200 magnification for each sample. All images of the liver sections were captured using a Nikon Digital Sight DS-U3 camera. Image analysis procedures were performed with Image J 1.49m (National Institutes of Health).
Point 4. Figure legend for Figure 1 should be revised.
Response: In the revised manuscript, we make changes in figure legends.
Page 7:
For panel (E) & panel (F) BALB/c male mice at 8 weeks of age were intraperitoneally injected with various amounts of HPS (0.2mg/kg or 1.0 mg/kg) or PBS 12h before and 24h and 48h after 100μl/kg CCl4 (1:100 diluted in corn coil) injection as indicated. (E) Mice serum were harvested and ALT levels detected at indicated time points after CCl4 injection. (n=5/group). (F) Liver tissues were fixed, sectioned and stained with H&E for histopathological and morphological analysis at indicated time points after CCl4 injection. Scale bar, 200μm. The part enclosed by a dashed line represents the necrotic area. Part of the necrotic area in the 24h group and 36h group is enlarged and displayed below. The percentage of necrotic area was quantitated using ImageJ software and values are the mean ± SD of five fields of measurements.
Point 5. Line 195: ‘phosphoralation’ should be changed to phosphorylation. Please correct other typos in manuscript as well.
Response: We are sorry for this mistake. In the revised manuscript, we correct these spelling errors.
Page 7:
Immunoblotting was performed to detected ERK phosphorylation.
Point 6. the manuscript needs minor work on grammar and spelling errors.
Response: We proofread the manuscript carefully, and correct grammar and spelling errors in the revised manuscript.
Round 2
Reviewer 1 Report
Authors responded for my questions and comments.